# What does the demographic profile of convicts tell us about social equity in India?

**Pranab Mukhopadhyay**[1]⊛*, **Shaila Desouza**[2]⊛, **Aparna P. Lolayekar**[2]⊛

**1** Goa Business School, Goa University, Goa, India, **2** Manohar Parrikar School of Law, Governance and Public Policy, Goa University, Goa, India

⊛ These authors contributed equally to this work.
* pm@unigoa.ac.in

## Abstract

Social equity is a critical component of social justice and is measured in multiple ways. Conventionally, researchers use literacy levels, workforce participation, political participation and representation, corporate presence, and demographic parity as indicators of social and economic equity. We add law enforcement outcomes to this literature by examining the demographic profiles of convicts in prisons in India for each state and compare it with that of the population of the respective state. To test whether entrenched social inequities have permeated into the law enforcement system, we use three indicators of social identity–religion, caste, and domicile–to create a social equity index (SEI). This is a composite index combining caste, religion and domicile similar in method to the Human Development Index which combines income, education and health. Our indicators are not considered in other popular development indices and is a conceptual innovation. Our paper innovates by combining prison data and census data at the state level over the two latest census rounds (2001 and 2011). We use a spatial panel analysis as well as a distributional dynamics approach to test for bias and transitions over time at the state level. We find that entrenched social hierarchies are mirrored in conviction outcomes and that social identities influence law enforcement. In contrast to earlier studies, we find that states that are conventionally considered to perform poorly in terms of economic and human development have done better in terms of social equity than economically advanced states.

## Introduction

Social identities have become increasingly important in the public sphere across the globe [1]. Ethnic, religious, and racial identities continue to dominate individual and social decision-making across the world [2, 3]. Even in countries that are multicultural and have witnessed waves of migration like the US, instances such as the death of George Floyd and the 'Black Lives Matter' movement remind us that issues of social equity, prejudice, and discrimination continue to determine social dynamics [4, 5] These events revive memories of Jim Crow laws that legalized racial segregation [6] and court rulings in the *Plessy vs Ferguson* case in the US [7]. Similarly, in Europe, racial divisions have resurfaced not only in public life [8] but also in

**Data Availability Statement:** Some of the data underlying the results presented in the study are available from a subscription source https://epwrfits.in/ and part of it is available in the public

domain. The sources have been indicated in the paper.

**Funding:** The authors received no specific funding for this work.

**Competing interests:** The authors have declared that no competing interests exist.

commercial sports [9]. While the resurgence of white supremacy in the US and Europe is alarming, ethnic conflicts have been equally contentious in Africa [10], Australia [11], and Asia [12], often escalating into 'hate crimes' [13].

Along with social hierarchies and religious identities, the emergence of the idea of the 'son of the soil' as an axis of identity has also been observed across the world. Studies have found an overrepresentation of immigrants in convictions in Nordic countries irrespective of their country of origin and proportion of population [14]. Political controversies involving immigrants are high on the political and policy agenda of many countries in Europe, and foreign migrants are consequently likely to be arrested, convicted, and imprisoned for violent, property, and drug crimes [15]. In Italy, immigration detention is seen as a mechanism to prevent crime and as an instrument of social defense [16]. In the US, racial/ethnic biases have influenced sentencing outcomes not only for blacks but also for Hispanics and other ethnic groups [17]. Bias in social and administrative systems impacts both the public and private spheres [18].

Evidently, the issue of social inequities persists in developed countries [19] despite their considerable advancement in terms of other social and economic indicators [20]. It is, therefore, unsurprising that the issue persists in less developed countries where basic amenities like water [21], energy [22], and health [23] remain inaccessible to large segments of the population, exacerbating existing social divisions. There have been attempts to understand and compare social discrimination across countries and over time [24] using international longitudinal datasets [25]. In this study, we focus on India, which has seen enormous growth after embracing a new economic model in the 1990s [26] and is soon expected to overtake China as the most populous country in the world [27]. India faces multiple development challenges that not only influence national outcomes [28] but also have international ramifications [29].

In India, apart from the global categories of class, color, religion, and ethnicity (including domicile [30]), there is an additional site of discrimination based on social hierarchy called caste [31, 32], which has similarities with social hierarchies in other parts of the world [33]. The caste system in India is a multidimensional, discriminatory, and hierarchical system where one's position in society is determined by one's birth into an endogamous community. It is distinguished by three characteristics: 1) separation, 2) division of labor and 3) hierarchy [34]. Anti-caste sentiments were first voiced by social reformers in pre-independent India [35] and feature prominently in the Constitution of India. Subsequently, laws were introduced to check discrimination, untouchability, and the criminalization of castes [36], and affirmative action in the form of reservations in higher education and public sector jobs was introduced. Some believe with great optimism that India's 2000-year-old caste system is fast breaking down and even likely to disappear in the near future [37]. There is some supporting evidence of this transition [38], but there is also a view that the caste system is being replaced by caste identities [38]. Either way, ground realities indicate that caste hierarchies continue to exist in India just as they do in other parts of the world [33]. Further, political parties in India have increasingly exploited caste identities to mobilize voters in most states in India [39, 40]. This trend has accelerated since the Mandal Commission proposed reservations for groups under the Other Backward Classes (OBC) category [41]. Individual political parties now promise benefits to castes that support them, leading to greater coalescing of caste identities and caste-based mobilization. Recently, 'Economically Weaker Sections' of the 'General' category were granted reservations by the ruling party to mobilize poor upper-caste households [42].

Similarly, studies on the evolution of religious conflicts in contemporary India from a historical perspective [43] suggest that the colonial state used it to deter the national movement. Post-Independence religious conflicts have been documented [44], and different explanations have been proposed to explain their occurrence [45, 46]. One path proposed to link religion and crime (conflict) is the income differential among communities, which is used as an

explanatory factor [47]. Just like caste, religion is used in India to mobilize political support [48] and establish social acceptance for majoritarian rule [49].

These categories are independent of each other. However, they overlap and intersect in the Indian context [50]. Therefore, caste, religion, and domicile are categories in which hierarchies and identities are intertwined. Caste hierarchies, for example, are known to exist not only among Hindus [33, 51], but also among other religions like Buddhists [52, 53], Christians [54, 55], Muslims [56, 57], among others.

It is widely recognized that the fundamental pillar of democracy–elections–often revolve around issues of religion, caste, and ethnicity [58, 59]. Elected governments mirror the opinions of majority groups, which also influences the allocation of public goods [60]. Social dominance gets ingrained in every walk of public and private life [61], including law enforcement [62].

In this context, two articles of the Indian Constitution emphasize the role of the state. Article 14 of the Constitution of India states that "The State shall not deny to any person equality before the law or the equal protection of the laws within the territory of India." Further, Article 15(1) states that "The State shall not discriminate against any citizen on grounds only of religion, race, caste, sex, place of birth or any of them." After almost 75 years of Independence, and over seven decades since the adoption of the Constitution of India, there is a need to examine how the state has performed in fulfilling its constitutional promise.

Contemporary concerns in India around domiciliary [63], religious [64], and caste [51] dynamics are reflected in research as well as public policy [65]. The Sachar Committee Report mentions that "a discriminatory attitude towards Muslims is felt widely" and that "police along with media, overplay the involvement of Muslims in violent activities and underplay the involvement of other groups or organizations" [65].

How does a society ascertain whether it is socially equitable to all sections of its population across time and regions? One means of assessment is analyzing prison statistics [66]. Studies in the US show that in 10 states, the proportion of black men sentenced to prison is 27–57 times higher than white men [67]. In India, a few studies have assessed bias in the administrative and justice system based on the race and ethnicity [68], gender [69], and religion of judges [69]. Social equity is considered an important goal of public administration, on par with economy and efficiency [62]. However, to the best of our knowledge, there are no studies in India that have examined the performance of the law enforcement system over time using national-level data, which is a research gap that needs to be addressed [70].

In this paper, we test for bias in the law enforcement system in India with respect to social equity using information on conviction rates and demographic details in a spatial framework. Accordingly, we pose three research questions: a) Is the law enforcement system in India socially equitable? b) Does social equity among states have a spatial dimension? and c) Has there been a transition in social equity among states over time? Our study focuses on the differences within each state and we compare the status of the most populous group with other groups in each state. We hypothesize that the proportion of convicts by category (religion, caste, and domicile) should not be significantly different from the proportion of the same category in the respective state's population. We further compare the extent of social equity with that of human development and gender equity. Our expectation is that education and higher incomes should lead to transparent, objective legal processes which are not influenced by conventional social hierarchies. Therefore, states/ regions that display better development indices should have overcome conventional social barriers reflected in the SEI. A state with higher HDI and GDI should reflect in reduced inequalities in social indicators.

We find that religion, caste, and domicile play a determining role in convictions across all states when law enforcement is assessed holistically. For all the social groups assessed–based on religion, caste, and domicile–there is a significant difference between the group's

proportion among convicts and the group's proportion in the population. The most populous group is not universally any one religion or caste but varies according to the demographic characteristics of each state. The human development index (HDI) and gender development index (GDI) both have a positive impact on the developed social equity index (SEI). Further, there is a (spatial) neighborhood effect on social equity. Interestingly, some of the Empowered Action Group (EAG) states, which comprise the eight states of Bihar, Chhattisgarh, Jharkhand, Madhya Pradesh, Orissa, Rajasthan, Uttarakhand, and Uttar Pradesh, which are considered development laggards, exhibit greater social equity than others that are considered better off by per capita income, HDI, and GDI measures. We propose a new measure of social equity that could complement existing development measures.

## Material and methods

### Data

This study examines the demographic profiles of convicts and compares them with the respective state's population, unlike other studies that have examined the judgments of courts [69] and the filing of police complaints [71]. The data for our study are drawn from four sources: a) the National Crime Records Bureau's (hereafter 'NCRB's') annual publication, *Prison Statistics India* (see https://ncrb.gov.in and S1 File), b) the Population Census of India (hereafter 'Census') (https://censusindia.gov.in), c) the Global Data Lab (hereafter 'GDL') [72, 73] and d) the Economic and Political Weekly Research Foundation database (hereafter EPWRF) (https://epwrfits.in/index.aspx). With regard to the demographic particulars of the convicts, we have considered their (1) religion, (2) social stratification (caste), and (3) domicile. The Census of India categories the population by religion [74] into 7 categories–Hindu, Muslim, Christian, Sikh, Buddhist, Jain and Others (which is a residual category). Similarly, the population is also categorised by caste [74] under three groups—SC, ST and Others. The last group includes the residual population that does not belong the marginal groups of SCs and STs. The Census also provides many details of the population for domicile characteristics which include birthplace and migration information [74].

NCRB did not provide a gender wise sub-classification therefore these numbers include both men as well as women convicts. We use HDI data and GDI data for India at the state level from the GDL for the relevant years (see S2 File). We use the per capita income (PCI) data from the EPWRF for the relevant years. We use secondary data that are available either free or by subscription in the public domain, and it is fully anonymized. As stated earlier, we have used data from 3 sources: 1) NCRB 2) EPWRF 3) Census of India.

Since data for this non-interventional study was obtained from published government sources and research foundations who had placed these datasets in anonymised form, there was no requirement to obtain institutional ethics approval. NCRB data is widely used by researchers on crime in India [75, 76]. The NCRB publications do not explicitly specify the nature of ethical compliance, however, they maintain primary ethical guidelines since their data is de-identified before release in the public domain and it is not possible to re-identifying respondents. The data collection exercise seems to record information that is routinely collected by law enforcement agencies and it is not clear from documentation available whether respondents would have a choice of consent. The EPWRF provides macroeconomic data which by its very nature provides de-identified individual information. The income dataset used from the EPWRF database would not allow re-identifying individual respondents. The data provided by the Census of India is governed by the Census Act, 1948. The protocols of ethical data collection and dissemination are as stipulated in the Census Act itself. The

information collected during the population Census is confidential and it is not even accessible to the courts of law. Therefore, confidentiality and anonymity is maintained.

One methodological novelty in our study is the combined use of PCI, census and GDL data with NCRB data for all states and union territories (UTs), including the National Capital Territory (NCT) of Delhi (i.e., 28 states, 6 UTs, and 1 NCT). However, data for states and UTs are not uniformly available. We have accordingly restricted our study to the states (except Arunachal Pradesh) and the NCT region for which complete data are available for all the variables.

## Methods

We used three methods to address each of the three questions that we posed earlier. To recapitulate, the first question examines social equity in the law enforcement system of each state. To address this question, we use gap analysis in combination with the SEI. The second question examines whether there is a spatial dimension to social equity in each state. To assess this, we use the spatial regression method [77, 78]. Finally, we examine whether there has been a change in social equity among the states. Here, we use the distribution dynamic approach [79, 80]. This method is preferred when non-parametric methods are required to study distributional changes over time without requiring a predictive model. We chose to not use regression analysis for this question as we are not proposing a predictive model for the outcome variable using causal factors. Further, the distribution dynamic approach is less restrictive and does not require the strong assumptions that regression analysis imposes on the model.

As a first step, we calculate the proportions of each demographic group among the convicts (DP) and the population (CN). The hypothesis we test is whether, *ceteris paribus*, all sections of the population are proportionately represented in legal outcomes, irrespective of religion, caste, and domicile status (see S2 File). Equality of means for paired samples is commonly used in the literature to test this kind of hypothesis [69].

### Gap analysis

First, we undertake a gap analysis for each demographic category as below (Eq 1):

$$
\begin{aligned}
Gap_{it} = &\ (\text{proportion of convicts belonging to the most populous group among the convicts} \\
&\ - \text{proportion of the most populous group in the population})
\end{aligned}
\tag{1}
$$

where, $I$ represents the state, and
  $t$ represents year

Gap analysis of consumption expenditure in India has been used to explain conflicts between social [81] and religious groups [47]. In contrast, we use a community's proportion in the population as an indicator of its relative social power for our gap analysis. The difference between the group's proportion among convicts and the group's proportion in the population is computed for each state. A group is stated to be most populous if its members comprise more than 50% of the population. We expect that a state would be socially equitable if this gap as measured in Eq 1 for all three categories (religion, caste, and domicile) is zero.

In the case of religion, all states except Punjab (Sikhs), Jammu & Kashmir (Muslims), and Meghalaya, Mizoram, and Nagaland (Christian) had a Hindu majority. For caste, all states except Meghalaya, Mizoram, and Nagaland (STs) had 'others' as the most populous caste group. For domicile, all states reported that the group born in the state was most populous over migrants from other states in India or abroad (see S3 File). This idea finds resonance in Sidgwick's equity principle in public finance for taxation, which asserts that "equally situated" groups should be "equally treated" [82]. The same can be extended to the realm of justice and

punishment. We further postulate that if $Gap_{it} > 0$, then it is treated as $Gap_{it} = 0$. The reason for this is that it indicates that the most populous group is not using its position in the social hierarchy to impose any form of majoritarianism on minority groups. Therefore, $Gap_{it}$ can only take values less than equal or to zero ($Gap_{it} \leq 0$) under our assumption.

## Religion equity index

We used the gap values to construct the religion equity index along the lines of the HDI [73] as below (Eq 2):

$$\text{Index}_{\text{ReligionEquity}} = (\text{Rel Gap}_{it} - \text{Rel min}_{it})/(\text{Rel max}_{it} - \text{Rel min}_{it}) \qquad (2)$$

For each state, we calculated the religion gap depending on the most populous religious group in that state as discussed above. In the case of religion, all states except Punjab (Sikhs), Jammu & Kashmir (Muslims), and Meghalaya, Mizoram, and Nagaland (Christian) had a Hindu majority.

Similarly, we create indices for caste and domicile (Eqs 3 and 4, respectively):

$$\text{Index}_{\text{CasteEquity}} = (\text{Caste Gap}_{it} - \text{Caste min}_{it})/(\text{Caste max}_{it} - \text{Caste min}_{it}) \qquad (3)$$

$$\text{Index}_{\text{DomicileEquity}} = (\text{Domicile Gap}_{it} - \text{Domicile min}_{it})/(\text{Domicile max}_{it} - \text{Domicile min}_{it}) \quad (4)$$

## Social equity index

The choice of variables and the assignment of weights in an index is often a debatable issue. Most often it is based on the subjective judgement of the author. Indices are often more easily understood and convey greater information than single indicators. This is the reason we rely on an index that conveys social equity. The theoretical framework that anchors this index is that of equi-proportionate representation in the population. In India, social groups are defined along many axes but most dominantly along caste, religion and domicile and are critical in public policy decisions. However, no known index uses these categories for policy analysis. By default, in the absence of a well-defined mechanism to assign weights to components of an index, equal weights are considered for the components [84]. The choice of these groups also avoids the problem of multicollinearity unlike other popular indices like HDI and GDI. There is no evidence to show that caste, religion and domicile are correlated. However, it is known that they will have inter-sectionalities. A system of caste may exist among all major religious groups in India. Similarly, different castes and religions will exist among those who are domiciled in a state or are migrants.

We used the three equity indexes (religion, caste, and domicile) to construct the SEI along the lines of the HDI [83], as below (Eq 5). This is done for each state in each year (2001 and 2011, individually) by taking a geometric mean of the three dimensions (characteristics). This is a replication of the HDI method as stated above.

$$SEI = (I_{religion} * I_{caste} * I_{domicile})^{1/3} \qquad (5)$$

After estimating the SEI, we explore two questions: First, is there a spatial dimension to social equity? Do neighboring states enjoy similar levels of social equity? Second, the development literature characterizes the EAG states as low achievers in terms of social and economic indicators [84]. Given that there is a well-recognized divergence in the incomes of states, especially after the structural reforms in India since 1990 [85, 86], how do the EAG states perform on the social equity front?

## Spatial regression method

To test for neighborhood effects, we use an econometric approach to understand the determinants of social equity. It is well-known in the literature that pooled ordinary least squares (OLS) models suffer from certain limitations [87, 88] (see S4 File). Panel data models are preferred to pooled OLS as they improve the efficiency of the estimates and control of omitted variable bias. However, as there is evidence of spatial dependence among states in India for socio-economic characteristics [89], it is advisable to use a modified version of the fixed effects model which controls for spatiality. To test for the presence of spatial dependence, we used the spatial autocorrelation (SAC) model [78]. This spatial panel model considers both endogenous interaction effects and interaction effects among the error terms.

We use a spatial panel model with a row normalized contiguity matrix to examine this question [90]. In Eq 6, the product of the weight matrix (W) and the dependent variable (SEI), WSEI, denotes the endogenous interaction effects of the SEI of neighboring states (Eq 6). We also test for interaction effects in the error terms by using 'Wu' to predict the stochastic error (Eq 7). The specific regression model tested is specified below:

$$Y_{it} = \alpha_{it} + \beta_1 X_{it} + \rho W Y_{it} + \mu_i + \eta_t + u_{it} \tag{6}$$

$$u_{it} = \lambda W u + e_{it} \tag{7}$$

where,

$Y$ = SEI

$X$ = HDI or GDI

$W$ = Spatial Weight Matrix (row normalized contiguity)

$e$ = Stochastic error term

$i$ = $i^{th}$ state

$t$ = year (2001 or 2011)

$\rho$ = estimate of interaction effect among SEI of states (spatial autoregressive coefficient)

$\mu$ = state-specific effects

$\eta$ = time-specific effects, and

$\lambda$ = estimate of the interaction effect among the error terms of states (spatial autocorrelation coefficient)

Following the work of [79], several studies [91–93] have pointed out the inadequacies of the regression approach in explaining the transitions of states over time. As an alternate, they propose a non-parametric approach, as discussed below.

## The distribution dynamics approach

Conventional regression analysis (cross-section, time series, and panel data estimations) rely on certain aspects of estimated conditional means but do not capture the distributional dynamics between or within regions [94]. The distribution dynamics approach studies the evolution of economies where the behavior of any economy is studied as the evolution of an entire distribution (see S5 File). This method reveals how one part of the distribution would behave with respect to another, and we use it to estimate the intra-period distribution change in the SEI. The distribution dynamics information is encoded in a transition-probability matrix. It is a square matrix that describes the probabilities of moving from one state to another in a dynamic system. Our matrix estimates the probabilities of mobility or persistence of a given region with regard to the SEI. A region, over a given time period, either remains in the same position or changes its position in the SEI. We present our results in the next section.

## Results

We present the summary statistics of the three social categories discussed above in Table 1. Our first objective was to check if the mean of the proportion of a group in the convict population (column C) and state population (column D) differ significantly by measuring the mean difference (column E). The t-test confirms whether the mean difference is significant or not (t-stat value is presented in column G and their corresponding p-value in column H).

For the caste groups, we find the differences to be significant between Scheduled Castes (SC) and 'Others' but not significant for Scheduled Tribes (ST). Importantly, for SCs, the group's share of convicts is more than its population share, but for the Others (higher castes), it is the opposite (the group's share of convicts is significantly lower than its population share). For the religious groups, we find the difference to be significant for Hindus, Muslims, and Sikhs but not significant for Christians and 'Others'. Hindus' share of convicts is significantly smaller than its population share, while the opposite is true for Muslims and Sikhs.

In the domicile groups, all three categories have significantly different means. The proportion of domicile convicts is smaller than the respective population proportion while the reverse is true for migrants from other states of India and other countries.

While this gives us a national-level picture, India is a diverse country with heterogeneity among states in terms of most populous castes and religious groups. Thus, we also examined the sub-national data. We next present the findings of the three indices created.

In the state-wise indices of SEI, we find that in 2001, Jammu & Kashmir (0.64) had the lowest equity for religion, Meghalaya (0.64) for caste, and Goa (0.35) for domicile (see Table 2). In the same year, Karnataka, Odisha, Manipur, and Mizoram (1) had the highest equity for religion, Assam, Delhi, Jammu & Kashmir, Madhya Pradesh, Manipur, and Sikkim (1) for caste, and Andhra Pradesh, Assam, Bihar, Chhattisgarh, Gujarat, Jharkhand, Karnataka, Madhya Pradesh, Maharashtra, Odisha, Punjab, Sikkim, Tamil Nadu, and Uttar Pradesh (1) for domicile.

For 2011, we find that Jammu & Kashmir (0.72) had the lowest equity for religion, Maharashtra (0.78) for caste, and Manipur (0.7) for domicile. In the same year, Goa, Jharkhand, Manipur, Mizoram, and Sikkim (1) had the highest equities for religion; Goa, Manipur,

**Table 1. Summary statistics and paired difference of means.**

| Category | Sub-category | Convicts (Mean) | Census (Mean) | Mean(Dif)* | Observations | T-stat | Pr(\|T\| > \|t\|) |
|---|---|---|---|---|---|---|---|
| A | B | C | D | E | F | G | H |
| Caste | SC | 19.25 | 13.16 | 6.08 | 59 | 6.26 | **0.0000** |
| | ST | 20.57 | 18.98 | 1.59 | 59. | 1.28 | 0.21 |
| | Others | 60.18 | 67.86 | -7.67 | 59 | -5.34 | **0.0000** |
| Religion | Hindu | 61.20 | 69.27 | -8.07 | 62 | -4.41 | **0.0000** |
| | Muslim | 16.78 | 11.52 | 5.26 | 62 | 4.27 | **0.00** |
| | Sikh | 4.41 | 2.97 | 1.44 | 62 | 2.69 | **0.01** |
| | Christian | 13.44 | 12.89 | 0.55 | 62 | 0.51 | 0.61 |
| | Others | 4.17 | 3.34 | 0.82 | 62 | 0.69 | 0.50 |
| Domicile (Born) | Domicile | 81.91 | 89.31 | -7.40 | 61 | -3.58 | **0.00** |
| | Other States | 12.38 | 9.62 | 2.76 | 61 | 1.71 | **0.09** |
| | Other Country | 5.71 | 1.07 | 4.64 | 61 | 2.48 | **0.02** |

Source: Authors' calculations based on NCRB and Census data.

* Mean(Dif) = Mean (Prop_convicts–Prop_census); SC = Scheduled Castes; ST = Scheduled Tribes

Note: Shaded cells of column H signifies significant difference and critical values less than 10%.

**Table 2. State-wise social indices of religion, caste, and domicile.**

| | Religious_index | | Caste_index | | Domicile_index | |
|---|---|---|---|---|---|---|
| **Year and States** | **2001** | **2011** | **2001** | **2011** | **2001** | **2011** |
| Andhra Pradesh | 0.92 | 0.86 | 0.83 | 0.91 | 1 | 1 |
| Assam | 0.96 | 0.99 | 1 | 0.83 | 1 | 0.97 |
| Bihar | 0.87 | 0.99 | 0.84 | 0.95 | 1 | 0.99 |
| Chhattisgarh | 0.89 | 0.92 | 0.82 | 0.89 | 1 | 1 |
| Delhi | 0.9 | 0.95 | 1 | 0.97 | 0.41 | 1 |
| Goa | 0.85 | 1 | 0.96 | 1 | 0.35 | 0.76 |
| Gujarat | 0.81 | 0.87 | 0.76 | 0.93 | 1 | 0.84 |
| Himachal Pradesh | 0.7 | 0.99 | 0.75 | 0.98 | 0.85 | 1 |
| Haryana | 0.82 | 0.96 | 0.95 | 0.9 | 0.97 | 1 |
| Jammu & Kashmir | 0.64 | 0.72 | 1 | 0.83 | 0.81 | 0.76 |
| Jharkhand | 0.83 | 1 | 0.93 | 0.99 | 1 | 1 |
| Karnataka | 1 | 0.89 | 0.95 | 0.9 | 1 | 1 |
| Kerala | 0.89 | 0.88 | 0.99 | 0.95 | 0.97 | 0.97 |
| Madhya Pradesh | 0.91 | 0.94 | 1 | 0.99 | 1 | 1 |
| Maharashtra | 0.88 | 0.82 | 0.81 | 0.78 | 1 | 0.95 |
| Manipur | 1 | 1 | 1 | 1 | 0.78 | 0.7 |
| Meghalaya | 0.8 | 0.95 | 0.64 | 0.8 | 0.69 | 0.72 |
| Mizoram | 1 | 1 | 0.98 | 1 | 0.99 | 0.99 |
| Nagaland | 0.66 | 0.77 | 0.87 | 0.94 | 0.61 | 0.94 |
| Odisha | 1 | 0.99 | 0.77 | 0.94 | 1 | 1 |
| Punjab | 0.86 | 0.95 | 0.79 | 0.81 | 1 | 1 |
| Rajasthan | 0.86 | 0.9 | 0.93 | 0.92 | 0.95 | 0.97 |
| Sikkim | 0.86 | 1 | 1 | 0.98 | 1 | 0.89 |
| Tamil Nadu | 0.86 | 0.8 | 0.75 | 0.87 | 1 | 1 |
| Tripura | 0.83 | 0.94 | 0.87 | 1 | 0.95 | 0.97 |
| Uttar Pradesh | 0.97 | 0.94 | 0.92 | 0.88 | 1 | 1 |
| Uttarakhand | 0.87 | 0.64 | 0.84 | 0.83 | 0.61 | 0.8 |
| West Bengal | 0.73 | 0.81 | 0.89 | 1 | 0.99 | 0.83 |

Source: Authors' calculations based on NCRB and Census data.

Mizoram, Tripura, and West Bengal (1) for caste; and Andhra Pradesh, Chhattisgarh, Delhi, Himachal Pradesh, Haryana, Jharkhand, Karnataka, MP, Odisha, Punjab, Tamil Nadu, and Uttar Pradesh (1) for domicile.

We next present our regression results from the panel (columns B and C) and spatial (SAC with contiguity matrix, columns D and E) models below (see Table 3). We find that in the (non-spatial) panel model, the beta coefficients for HDI (0.397) and GDI (0.644) are significant. The same is true for the beta coefficient for the spatial panel for HDI (0.309) and GDI (0.537), confirming a positive relationship between the SEI and its covariates HDI and GDI. We also find that there is a significant difference between the beta values estimated from the panel regression and the spatial panel regression for both the HDI and GDI; the t-test difference of means with unequal variances has p-values of 0.0024 for HDI and 0.0076 for GDI. This justifies the use of spatial panel methods, since there is a possibility that the non-spatial models may overestimate the beta values [78].

The results of the spatial regression provide us with additional items, namely rho (($\rho$), 0.304 and 0.427) and lambda (($\lambda$), –0.302 and –0.528), for HDI and GDI, respectively. We find that

**Table 3. Regression results for panel and spatial panel fixed effects (with robust standard errors for robust regression).**

| SEI | Panel regression | | SAC row model (SEI) | |
|---|---|---|---|---|
| | HDI | GDI | HDI | GDI |
| A | B | C | D | E |
| beta | 0.397 | 0.644 | 0.309 | 0.537 |
| se (robust) | 0.18 | 0.27 | 0.166 | 0.206 |
| t-stats | 2.21 | 2.35 | 1.86 | 2.61 |
| p-value | **0.034** | **0.024** | **0.063** | **0.009** |
| rho (ρ) | | | 0.304 | 0.427 |
| p-value | | | **0.175** | **0.019** |
| Lamda (λ) | | | -0.302 | -0.528 |
| p-value | | | **0.107** | **0.007** |
| Log likelihood (pseudo) | | | 131.6 | 129.6 |
| R-square (within) | 0.14 | 0.06 | 0.1629 | 0.1314 |
| Number of obs | 74 | 74 | 74 | 74 |
| Groups | | | 37 | 37 |
| Year | | | 2 | 2 |
| Mean of FE* | 0.65 | 0.375 | 0.4508 | 0.1058 |

Note: Minmax and Spectral results are similar (not reported).

*Mean of FE is reported as constant in non-spatial model results

Source: Authors' calculations.

in the spatial regression, the estimates 'ρ' is positive and 'λ' is negative and significant for GDI (col E). This suggests that the SEI of any state, in addition to being influenced by the state's own GDI, is also influenced by the neighbor's SEI. However, in the case of HDI, the 'ρ' and 'λ' coefficients have the same signs as the GDI regression but are not significant.

## Analysis

We now examine the trajectory of each state with respect to their SEIs over the decade from 2001 to 2011 using the transition matrix (see Table 4). We use a $4 \times 4$ transition probability matrix to display the classification of states using a discretized spaces for SEI (ranging from 0.66 to 1 for 2001 and 0.75 to 1 for 2011). The bands that we used are based on the range of values in each year (2001 for columns and 2011 for rows). Row (and column) 1 considers the states with SEI values of less than 0.75, row (and column) 2 has SEIs that range between 0.75 and 0.85, row (and column) 3 has states with SEIs between 0.85 to 0.95, while row (and column) 4 has the states with the highest SEIs of above 0.95. Each cell in the matrix locates the states according to their relative SEI in the starting and ending year.

We find that fifteen states have moved from their initial period band location, wherein eleven states moved above the diagonal, and four states moved below the diagonal. Among the states that had low starting SEIs, Delhi and Haryana have shown remarkable improvement by moving three steps ahead, while Goa has shown a two-step transition, with Nagaland showing a one-step transition.

In the next quartile of SEIs, Bihar and Tripura have moved two cells above, and Chhattisgarh and Punjab have moved one cell above the diagonal. States like Jharkhand, Odisha, and Sikkim started with a higher SEI, and all moved a band above the diagonal in the terminal year. In contrast, Manipur started with a high SEI but moved one band below the diagonal. Similarly, Assam, Uttar Pradesh, and Karnataka initially had a high SEI but moved one band

**Table 4. Transition probability matrix for SEI (during 2001–2011).**

| | | 2011 | | | | Number of states |
|---|---|---|---|---|---|---|
| | | States ending in the first quartile | States ending in the second quartile | States ending in the third quartile | States ending in the fourth quartile | |
| 2001 | States starting in the first quartile | J&K, Uttarakhand, Meghalaya | Nagaland | Goa | Delhi, Haryana | 7 |
| | States starting in the second quartile | 0 | Gujarat, Tamil Nadu West Bengal, Maharashtra | Chhattisgarh, Punjab | Bihar, Tripura | 8 |
| | States starting in the third quartile | 0 | Manipur | Andhra Pradesh, Kerala, Rajasthan, Himachal Pradesh | Jharkhand, Sikkim, Odisha | 8 |
| | States starting in the fourth quartile | 0 | 0 | Assam, Uttar Pradesh, Karnataka | Madhya Pradesh, Mizoram | 5 |

Source: Authors' calculations.

below diagonally in the terminal year. States with low initial SEIs like Jammu & Kashmir, Meghalaya, and Uttarakhand were persistent at the diagonal in the terminal year. Similarly, Gujarat, Tamil Nadu, West Bengal, and Maharashtra in the next category of SEI maintained their positions on the diagonal. Andhra Pradesh, Kerala, Rajasthan, and Himachal Pradesh, which had high SEIs, also were persistent on the diagonal in the terminal year. Madhya Pradesh and Mizoram, which had the highest SEIs, maintained it in the terminal year. The EAG states (as indicated in the color blue) have outperformed the high-income states with respect to SEI.

Is the performance of the states in terms of SEI comparable to their PCI performance?

We use a 4 × 4 transition probability matrix to display the classification of states using a discretized space for PCI. For 2001, the PCI values ranged from 0.33 to 2.36, and for 2011, from 0.56 to 3.99 (see Table 5). The bands that we have used are based on the range of values in each year (2001 for rows and 2011 for columns). Row (and column) 1 considers the states that are in quartile 1, with PCI values that are less than 0.59 for 2001 and less than 0.95 for 2011. Row (and column) 2 has states that have PCIs that range between 0.59 and 0.92 for 2001, and states that have PCIs that range between 0.95 and 1.52 for 2011. Row (and column) 3 has states that have PCIs that range between 0.92 and 1.04 for 2001 and states that have PCIs that range

**Table 5. Transition probability matrix for PCI at 2011–12 constant prices (during 2001–2011).**

| | | 2011 | | | | Number of states |
|---|---|---|---|---|---|---|
| | | States ending in the first quartile | States ending in the second quartile | States ending in the third quartile | States ending in the fourth quartile | |
| 2001 | States starting in the first quartile | Bihar, Odisha, Jharkhand, Uttar Pradesh | Assam, J&K, Madhya Pradesh, Manipur, Meghalaya, Rajasthan, Chhattisgarh | Andhra Pradesh, Karnataka, Nagaland, Sikkim, Tripura, West Bengal, Uttarakhand | | 18 |
| | States starting in the second quartile | 0 | 0 | Gujarat, Kerala, Mizoram, Tamil Nadu | Himachal Pradesh, Maharashtra | 6 |
| | States starting in the third quartile | 0 | 0 | 0 | Haryana, Punjab | 2 |
| | States starting in the fourth quartile | 0 | 0 | 0 | Delhi, Goa | 2 |

Source: Authors' calculations based on *https://epwrfits.in/SDPTreeViewData.aspx*

between 1.52 and 1.71 for 2011. Row (and column) 4 has the states with the highest PCIs of above 1.04 for 2001 and 1.71 for 2011.

In terms of PCI, we find that 22 states have moved above the diagonal and 6 states maintained their position on the diagonal (see Table 5). Among the EAG category (as indicated in color red in the Table) Bihar, Jharkhand, Uttar Pradesh, and Odisha started with low PCIs in 2001 and ended with low PCIs in the terminal period. Madhya Pradesh, Manipur, Chhattisgarh, and Manipur (states with low PCIs in 2001) moved one band ahead. Andhra Pradesh, Karnataka, Nagaland, Sikkim, Tripura, West Bengal, and Uttarakhand started with low PCIs and moved two bands ahead. Himachal Pradesh and Maharashtra made a significant improvement by moving two bands above their original band. Delhi and Goa maintained their higher PCIs even in 2011.

Conventionally, the development debate in India posits that the southern states outperform their northern counterparts [95, 96]. We test this claim by examining the spatial transition using choropleth maps, which help us to visualize the GDI, HDI, and SEI spatially (Fig 1). The maps are color-coded into four quartiles. The darkest shade of blue indicates the highest values (of GDI, HDI, and SEI), while the color red indicates the lowest. With regard to GDI, most of

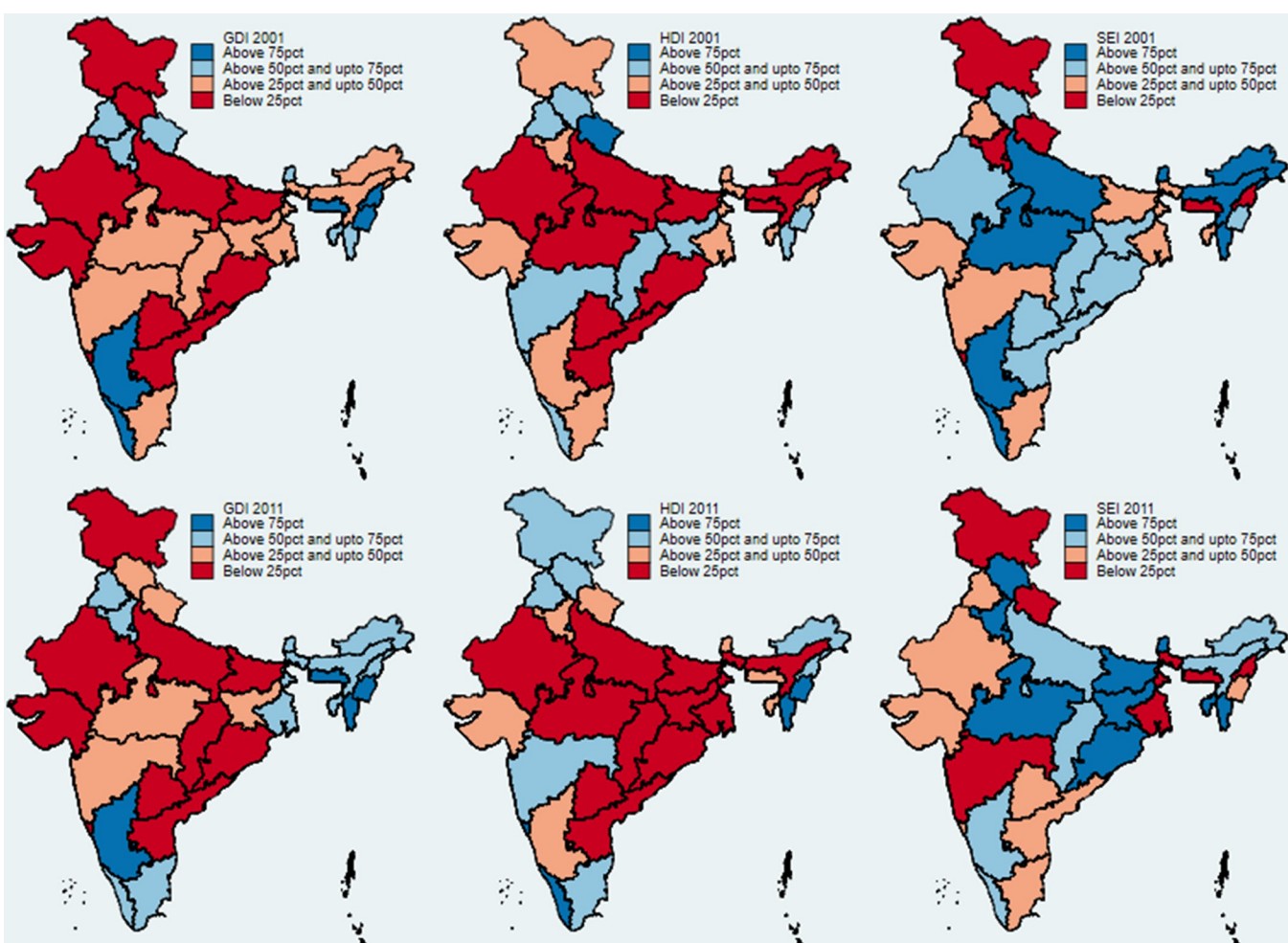

**Fig 1. State-wise relative GDI, HDI, and SEI (2001 and 2011) in India.** Source: Authors' calculations based on [97]. Attribution: "Parliamentary Constituencies Maps are provided by Data{Meet} Community Maps Project. It is made available under the Creative Commons Attribution 2.5 India".

the states in the central, east, and west region of India had low GDIs in 2001. In the southern region, Tamil Nadu moved from the second to the third quartile, while Kerala dropped from the fourth quartile to the third. The northeastern states showed a significant improvement. With regard to HDI, the central Indian states are a cause for concern. Almost all the states are in the first quartile in 2011. In the deep south, both Tamil Nadu and Kerala have improved their relative HDI performance. When we turn to the spatial SEI map, we find that the EAG states have performed relatively better compared to their developed counterparts. This presents a contrast to the GDI and HDI spatial maps for India. Both in 2001 and 2011, the performance of the EAG states with respect to SEI has been far better than for HDI and GDI.

Goa, Delhi, Meghalaya, Nagaland, Uttarakhand, Jammu & Kashmir, and Haryana had low SEIs in 2001. Only Delhi and Haryana showed remarkable improvement in 2011, while Jammu & Kashmir and Uttarakhand have maintained their SEI. Maharashtra showed a decline in SEI in 2011. A similar case is observed in Karnataka, Uttar Pradesh, and Assam.

## Discussion

Our study addresses the knowledge gap in the empirical literature on social equity by assessing data on the outcomes of the law enforcement system in India. We posed three questions to examine social equity in law enforcement: whether it is equitable, whether it has a spatial dimension, and whether there is a change over time (transition). We have found that convictions across states have not been socially equitable. We also find evidence that there is a spatial dependence between states in terms of social equity. Over time, there has been a transition in social equity– 4 states fell behind their initial level and 15 states moved above their initial level. The rest remained in the same band in the initial and terminal years of study.

The values and principles of justice and betterment of society necessitate social equity [98]. In addition to being a constitutional mandate in India and many parts of the world, social equity feature prominently in the global Sustainable Development Goals (SDG), namely, the goals pertaining to gender equality (SDG5), reducing inequalities (SDG10), and peace, justice, and strong institutions (SDG16) [99]. It is, therefore, important that we examine the performance of the state apparatus in delivering on these constitutional requirements and meeting these global goals. These goals are not located in a vacuous space–inequalities and racial and social biases pervade and affect lives across the world.

We use an innovative approach that combines data from a nationwide demographic database of convicts (NCRB) and the Census of India and GDL to perform a broad-based assessment of social equity and the law enforcement system. Our assessment starts by undertaking a gap analysis of the proportion of particular social categories among convicts in prisons and the respective state. Our gap analysis suggests that for all social groups (religion, caste, and domicile), there is a significant difference between the groups' proportion among convicts and the population. Whichever group (among these three categories of social identity) was most populous in the state population showed a lower proportion among convicts in almost all states and categories in both 2001 and 2011. It is important to note that the most populous religious group or caste is not constant across the country but varies according to the demographic characteristics of each state.

Based on the gap analysis, we then constructed an SEI based on three aspects of social identity–religion, caste, and domicile. These, as the literature suggests, are important sites of social bias. The SEI was regressed against HDI and GDI (separately) to check if states that have a high HDI or GDI have a better SEI. Our regression results (panel data with fixed effects) suggest that both HDI and GDI have a positive impact on SEI in both non-spatial and spatial frameworks. The spatial (auto-correlation) regression results inform us that while HDI was

not significant (in the spatial framework), GDI continues to be positive and significant in determining the SEI. The spatial analysis further reveals that each state's SEI is impacted by its neighbor's SEI. This suggests that there are spill-over effects between states and their neighbors. Since regression models use restrictive assumptions and ignore intra-distribution mobility, we have also used the distribution dynamics approach to examine the transitions of states over the decade 2001–2011 in terms of SEI.

Some earlier studies have found no evidence of bias in the Indian legal system. The difference between these findings and ours is that we have examined the law enforcement system holistically from the point of registration of a complaint to the endpoint where the judicial system convicts the accused or dismisses the accusation. We provide evidence that at a systemic level, issues of religion, caste, and domicile determine convictions.

Evidence suggests that the judicial process is subject to potential bias that can subvert the social equity project, from the filing of a complaint to follow-up processes including registration, investigation, placing the matter in the judicial system, provision of evidence during the trial, and arguing the case before a judge. While a number of these issues have been recognized and policy efforts have been made to rectify them, there remain substantial challenges. Our findings point to some of the challenges to the social equity project and have significant academic as well as public policy implications.

The expectation of the social development process is that education and higher incomes should lead to transparent, objective administrative and legal processes that are not influenced by conventional social hierarchies. Therefore, states/ regions that display better development indices should have been able to overcome conventional social barriers that are presumably reflected in the SEI. A state with higher HDI and GDI should show reduced inequalities in religion, caste and domicile.

## Conclusion

The literature on human wellbeing finds income alone to be an inadequate indicator of development. The UN uses a combination of income, health, and education indicators to track development over three decades. In the Indian context, these ideas evolved even earlier when states were classified into groups like EAG and non-EAG. However, to the best of our knowledge, ours is the first attempt to use identities to assess social equity. Our method allows us to examine how the states classified as EAG perform in terms of social equity. Surprisingly, all the EAG states performed well on the SEI, being placed in the higher bands in both the initial year and the terminal year. Our estimates of crime rates in EAG states in 2001 and 2011 were lower than that in non-EAG states (S6 Table in S6 File). In contrast to [84], who found that EAG states continue to be socially backward and economically diverging in PCI [89, 93], we find that these so-called EAG states actually exhibit greater social equity than many others that are considered better off by PCI, HDI, and GDI measures. While Bihar, Odisha, and Jharkhand actually improved their SEI, Chhattisgarh, Rajasthan, and Madhya Pradesh maintained their bands between 2001 and 2011. Only Uttar Pradesh dropped below the diagonal by one band. In terms of PCI, the lowest-performing states were Bihar, Madhya Pradesh, Manipur, Odisha, Uttar Pradesh, Chhattisgarh, and Jharkhand. In sharp contrast, Goa and Delhi, the states with the highest PCI, were poor performers in terms of SEI in 2001. Our findings provide an opportunity to expand the development debate using identities as social equity indicators. As the ongoing social upheaval across the world suggests, identities play a significant role in determining human wellbeing. We provide a pathway to measure social equity at the sub-national level and a mechanism to incorporate it into existing development measures.

One limitation of this study is that it has treated caste, religion and domicile as independent characteristics. We are aware that there are intersectional overlaps between them. However, we feel that our results would be robust despite the intersectionality and future research in this area could explore these aspects in further detail.

## Supporting information

**S1 File.**
(DOCX)

**S2 File. T-test.**
(DOCX)

**S3 File.**
(DOCX)

**S4 File.**
(DOCX)

**S5 File.**
(DOCX)

**S6 File. S6 Table.** Summary statistics of average crime rates by different category of states.
(DOCX)

## Author Contributions

**Conceptualization:** Pranab Mukhopadhyay, Shaila Desouza, Aparna P. Lolayekar.

**Data curation:** Aparna P. Lolayekar.

**Formal analysis:** Pranab Mukhopadhyay, Shaila Desouza, Aparna P. Lolayekar.

**Methodology:** Pranab Mukhopadhyay, Shaila Desouza, Aparna P. Lolayekar.

**Writing – original draft:** Pranab Mukhopadhyay, Shaila Desouza, Aparna P. Lolayekar.

**Writing – review & editing:** Pranab Mukhopadhyay, Shaila Desouza, Aparna P. Lolayekar.

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
