## [Decision Letter · Decision Letter 0]

28 Apr 2022

PONE-D-21-38262What does the demographic profile of convicts tell us about social equity in India?PLOS ONE

Dear Dr.Pranab Mukhopadhyay,

Thank you for submitting your manuscript to PLOS ONE. After careful consideration, we feel that it has merit but does not fully meet PLOS ONE’s publication criteria as it currently stands. Therefore, we invite you to submit a revised version of the manuscript that addresses the points raised during the review process.

MAJOR REVISIONS

We look forward to receiving your revised manuscript.

Kind regards,

Muhammad Ishtiaq

Academic Editor

PLOS ONE

Journal Requirements:

3. We note that Figures 1 in your submission contain [map/satellite] images which may be copyrighted. All PLOS content is published under the Creative Commons Attribution License (CC BY 4.0), which means that the manuscript, images, and Supporting Information files will be freely available online, and any third party is permitted to access, download, copy, distribute, and use these materials in any way, even commercially, with proper attribution. For these reasons, we cannot publish previously copyrighted maps or satellite images created using proprietary data, such as Google software (Google Maps, Street View, and Earth). For more information, see our copyright guidelines: http://journals.plos.org/plosone/s/licenses-and-copyright.

   1. You may seek permission from the original copyright holder of Figures 1 to publish the content specifically under the CC BY 4.0 license.  

Maps at the CIA (public domain): https://www.cia.gov/library/publications/the-world- factbook/index.html and https://www.cia.gov/library/publications/cia-maps-publications/index.html

Additional Editor Comments (if provided):

Please revise the paper as per suggestions of the reviewers's and resubmit it.

Reviewers' comments:

Reviewer's Responses to Questions

**Comments to the Author**

1. Is the manuscript technically sound, and do the data support the conclusions?

Reviewer #1: Partly

Reviewer #2: Yes

2. Has the statistical analysis been performed appropriately and rigorously? 

Reviewer #1: No

Reviewer #2: I Don't Know

3. Have the authors made all data underlying the findings in their manuscript fully available?

Reviewer #1: No

Reviewer #2: No

4. Is the manuscript presented in an intelligible fashion and written in standard English?

Reviewer #1: Yes

Reviewer #2: No

5. Review Comments to the Author

Reviewer #1: Author needs to improve article more with justified description of the article findings clearly. Abstract should be concise and clearly elaborate. Introduction section needs improvement and recent reference add. Results should justify with clear indications

Reviewer #2: This is an innovative paper in its approach to a perennial issue globally and India, the context needs to be revised and hypothesis rearticlauted. The background should be recast to explain more lovally relevant social factors that may account for social inequitiy India

Please review comments for added feedback

6. PLOS authors have the option to publish the peer review history of their article (what does this mean?). If published, this will include your full peer review and any attached files.

Reviewer #1: No

Reviewer #2: **Yes: **Dr Danish Ahmad

---

## [Author Response · Author response to Decision Letter 0]

21 Sep 2022

We have added a detailed response to the reviewers as a separate file.

---

## [Decision Letter · Decision Letter 1]

28 Apr 2023

PONE-D-21-38262R1What does the demographic profile of convicts tell us about social equity in India?PLOS ONE

Dear Dr. Pranab Muphopadhyay,

Thank you for submitting your manuscript to PLOS ONE. After careful consideration, we feel that it has merit but does not fully meet PLOS ONE’s publication criteria as it currently stands. Therefore, we invite you to submit a revised version of the manuscript that addresses the points raised during the review process.

I have attached my comments as attachments.

We look forward to receiving your revised manuscript.

Kind regards,

Gouranga Lal Dasvarma, PhD

Academic Editor

PLOS ONE

Additional Editor Comments:

I was asked to handle this manuscript PONE-D-21-38262R1, when the Academic Editor initially handling the same became unavailable after the two required reviews and subsequent revisions of the manuscript. After your revised manuscript was reassigned to me I was given the choice to have a third reviewer go through your revised manuscript. However, after eight reviewers declined my invitation to review your manuscript I decided to go through the revised manuscript myself and found that it still needs revisions. I have attached my comments together with your revised manuscript (in track changes) as separate attachments. Please address the comments and submit another revision. Thank you.

Reviewers' comments:

Reviewer's Responses to Questions

**Comments to the Author**

1. If the authors have adequately addressed your comments raised in a previous round of review and you feel that this manuscript is now acceptable for publication, you may indicate that here to bypass the “Comments to the Author” section, enter your conflict of interest statement in the “Confidential to Editor” section, and submit your "Accept" recommendation.

Reviewer #1: All comments have been addressed

Reviewer #2: All comments have been addressed

2. Is the manuscript technically sound, and do the data support the conclusions?

Reviewer #1: Yes

Reviewer #2: (No Response)

3. Has the statistical analysis been performed appropriately and rigorously? 

Reviewer #1: I Don't Know

Reviewer #2: I Don't Know

4. Have the authors made all data underlying the findings in their manuscript fully available?

Reviewer #1: Yes

Reviewer #2: Yes

5. Is the manuscript presented in an intelligible fashion and written in standard English?

Reviewer #1: Yes

Reviewer #2: Yes

6. Review Comments to the Author

Reviewer #1: Author has been addressed all the assign corrections in previous review report. Now article is suitable for acceptance in this journal after good formatting and setting

Reviewer #2: Thank you for submittting a revised paper, I am happy to recommend it for publication.However I will add a line of caution regarding the introduction and let the editor decide if the introduction(para's 1and 2) is presented in accordance to PloS one guidance to authors for contemporary sensitivetopics.

7. PLOS authors have the option to publish the peer review history of their article (what does this mean?). If published, this will include your full peer review and any attached files.

Reviewer #1: **Yes: **Dr. Tanveer Hussain

Reviewer #2: **Yes: **DR Danish Ahmad,MBBS,MSc,PhD,MNAMS,IP-FPH

---

## [Author Response · Author response to Decision Letter 1]

12 Jun 2023

Reviewer's Comments and author responses (point wise)

We thank the reviewer for the extensive comments. Please see our response (in blue) point wise for all comments.

Line 33 SEI. Is this a composite index? If so, what are the components of the index (such as index of caste, index of religion and index of domicile)? 

We thank the reviewer for this comment. We have updated the text accordingly. 

Lines 39-42; 181; 290; 292; 294;446; 462; 486; 489; 574-576; 581. The acronym "BIMARU" (standing for Bihar, Madhya Pradesh, Rajasthan and Uttar Pradesh) was coined by the late demographer Ashish Bose in the 1980s to highlight the lack of progress of these states in family planning, control of population growth and general socio-economic development. The acronym appeared to be apt at the time, it was first used because the word bimaru in Hindi also means sickly, although its use was not without criticism. However, this acronym appears not to be in vogue anymore. Therefore, please remove the acronym BIMARU from the manuscript and use the more recent terminology EAG (Empowered Action Group) states, which comprise the eight states of Bihar, Chhattisgarh, Jharkhand, Madhya Pradesh, Orissa, Rajasthan, Uttaranchal, and Uttar Pradesh. https://www.careers360.com/eag-states-full-form

We thank the reviewer for this comment. We have updated the text with the term EAG wherever applicable.

Lines 117-123. Discrimination on the basis of religion may also be confounded by socio-economic status. A religious group which is discriminated against may be poorer in literacy/education and economic status. Do you acknowledge this? 

We agree with the view of the reviewer. We have changed the text in the paper accordingly.

Lines 163-165. Were the convicts you have studied all males, or were there females included? 

This is an important query. Our data from NCRB did not provide a gender-wise classification. The data includes both men as well as women convicts.

Are discriminations based on caste, religion and domicile the same for men and women? Were the convicts you have studied all males, or were there females included? 

This, too, is an important point of query. Since NCRB provides gender-based data of the kind we have used in this paper, we could not combine gender and religious (or caste) characteristics to examine differential effects.

Lines 166-167, Question b). This question needs clarification/elaboration. Are you looking at differentials in social equity by state or are you looking at differentials in social equity within states? 

This is a valid point of reflection. We are studying the differences within each state. We compare the status of the "most populous" group with other groups in the state. Accordingly, the text in the paper has been changed.

Line 177. It is not clear what you mean by "dominant" group. Dominant in what sense? In fact, the entire sentence needs clarification. 

We thank the reviewer for this comment. Although we had a notion of dominance in mind, we realised that this could mean multiple things to readers. What we had in mind was simple dominance in numbers (proportions). We have accordingly changed the text in the paper to replace "dominant" with the word "most populous."

Lines 178-180. Both the HDI and GDI are available in "inequality-adjusted" form. But these indices comprise scores on life expectancy, education and income, and not religion, caste or domicile. One can argue that indices like HDI or GDI (or their inequality-adjusted forms) are, in fact, influenced by inequalities among religion, caste and domicile 

We agree with the reviewer. One of the reasons for undertaking this study was to inform the debate on HDI and GDI of factors that were influencing inequalities that went beyond the identifiers of inequality being used in the HDI and GDI.

Regarding causality, we believe education and higher incomes should lead to transparent, objective legal processes that are not influenced by conventional social hierarchies. This is the expectation of development processes. Therefore, states/ regions that display better development indices should have been able to overcome conventional social barriers that are presumably reflected in the SEI. A state with higher HDI and GDI should show reduced inequalities in religion, caste and domicile. The text in the paper has accordingly been updated. 

Lines 178-179. Please explain how, because your SEI is based on different characteristics. It is difficult to see how HDI and GDI would have any impact on the newly constructed SEI or vice versa. 

This a valid request. However, we feel that HDI and GDI would influence SEI because economic and human development is expected to reduce social inequity. Extensive literature, including the premise of multiple SDGs, suggests this. We have elaborated on this in the previous response (to the comment above). In addition, we have updated the text in the revised paper.

Lines 180-181. Are the overall crime rates of the BIMARU/EAG states higher, or lower than those of the non-BIMARU/EAG states? 

We thank the reviewer for this query and have updated the text with additional information on crime rates in different types of states.

Lines 199-202. Data that are in public domain also need Ethics clearance in the first place to collect (example, Demographic and Health Survey data). Are you aware of any ethics clearance for the data you have used, and state what kind of ethic clearance(s) have been obtained? 

We have updated the information available on ethics clearance in our paper. Our data is freely available in the public domain, and we have updated the text with relevant information.

Lines 203-204. The use of different datasets is common in many studies with each study using its own combination of data sets, but it cannot be called an "innovation"! An innovation may be in an innovative data collection or in the methodology. 

We accept the comment of the reviewer and have altered the text accordingly. We have replaced the word innovation with novelty. We hope this is acceptable to the reviewer.

Line 211. The first question examines social equity in the law enforcement system of each state. 

We accept the comment of the reviewer and have altered the text accordingly. 

Line 213. Please correct to: Spatial dimension to social equity in each state. 

We accept the comment of the reviewer and have altered the text accordingly. It may be noted that the social equity issue is addressed in each state while the spatial dimension is examined between states.

Line 214. Please correct to: Change in social equity among the states. 

We accept the comment of the reviewer and have altered the text accordingly. 

Line 218. "overcomes" or does not require? 

We accept the suggestion and have altered the text accordingly.

Line 230. What is the dominant" group? 

We acknowledge that "dominant" group is an ambiguous group. What we had in mind was the "most populous" group, which is just the largest cluster by population proportion. We have replaced all references to the dominant group with this term.

Line230. The formula, as it is worded does not take the difference (gap) between similar things of two groups. The wording of first expression of this definition is misleading, as it would imply proportions of convicts (and non-convicts in the dominant group). Do you mean "proportion of the dominant group among the convicts"? Please check that you have used the correct proportion in the first expression of the right-hand side of the formula.

The reviewer is correct. We accepted the suggestions and accordingly used the phrase "proportion of the most populous group among the convicts" in the revised text.

Line 231. What is the "dominant" group? 

Please see our response in query reference to line 230 (above)

Lines 239-240. This appears to be correct. Please modify the wording of your Equation 1 accordingly.

The reviewer is correct. We accept the suggestion and accordingly have updated the revised text.

Line 240. What is the "dominant" group? 

Please see our response in query reference to lines 230 and 231 (above)

Line 242. Do you mean the gap, as purported to be shown in Equation 1? 

Yes. Thank you for pointing this out. We accept the suggestion.

Lines 244-246. The "Others" category is an amorphous or undefined group. How can any conclusion or policy recommendation be made for an undefined group? I suggest you consider the next proportionately largest group in this instance. 

We have added further details on the Census categories by religion, caste and domicile in the revised paper. There has been a longstanding demand by academics and political representatives to hold a detailed caste census. Still, the caste data is only available under these categories as specified in the 2001 and 2011 household schedules.

In the caste group, while "Others" may seem amorphous, many researchers and policy-makers feel that the "Others" includes all individuals who do not belong to the marginalised groups like SCs and STs and, therefore, adequate for many immediate policy needs. 

Lines 246-248. Have you considered the possibility that the group "born in the state" could comprise a mixture of religions and castes? What would that imply for your results and conclusions? 

We acknowledge the existence of the possibility the reviewer points to. Our response is similar here as in an earlier comment. 

These categories are independent of each other. However, they overlap and intersect in the Indian context [50]. Therefore, caste, religion, and domicile are categories in which hierarchies and identities are intertwined. Caste hierarchies, for example, are known to exist not only among Hindus [33,51] but also among other religions like Buddhists [52,53], Christians [54,55], and Muslims [56,57], among others. Our results would be robust to these overlaps. 

Future research could explore these intersections, and further sub-categorisation would add details to our findings without contradicting them."

Lines 253-254. Please add the words "under our assumption" 

We thank the reviewer for this suggestion. We have revised the text in the paper.

Lines 256-266. Please elaborate how Rel Gapit, Rel minit etc are calculated. What have you assumed for Rel max it, and why? Please elaborate the same also for caste and domicile.

In calculating the various indices (or scores) for HDI, standard maximum and minimum values of life expectancy, education and per capita income are taken. What maxima and minima have you assumed for the gaps in religion, caste and domicile and why have you assumed them so? 

As the first step, we calculated the proportion of convicts and population by each group (by religion, caste and domicile) for all states. Then we calculated the most populous category (religion, caste and domicile) among the convicts and each state's population. Most states have Hinduism as the most populous religion, except Punjab (Sikh), Jammu and Kashmir (Muslim), while Meghalaya, Mizoram and Nagaland have Christian populations. For each state, we calculated the religion gap depending on the most populous religion in that state. Similarly, the caste and domicile gap was calculated. For the gap analysis, the gap is the difference between the group's proportion among convicts and the group's proportion in the population (by religion, caste and domicile). The maximum values are set to the observed maximum values of the most populous groups (of religion, caste and domicile) in 2001 and 2011. The minimum value is set to zero. The three gap values were then used to construct respective indices. These three indices were then multiplied to obtain the geometric mean of SEI.

We have stated this in the main text, and so this text was not updated. However, if the reviewer feels that this needs elaboration, we would be agreeable to do so. "

Lines 268-272. The United Nations assigns equal weights to each of the three human development indices with the implicit assumption that each contributes equally to HDI. How true is that for the three indices of social equity in your study? There needs to be a discussion about assigning equal weights to the indices of caste, religion and domicile. Further, as I have mentioned earlier, domicile is an amorphous indicator in which caste and religion may be mixed up. This raises a further question similar to that of multicollinearity as in multiple regression analysis. Please provide a satisfactory discussion on this matter. 

The choice of variables and the assignment of weights in an index is often debatable as it is usually based on the author's judgement. Indices are often more easily understood and convey greater information than single indicators. This is the reason we rely on an index that conveys social equity. The theoretical framework anchors this index is equiproportionate representation in the population. In India, social categories are clearly defined along many axes but most dominantly on caste, religion and domicile. These groups are not part of any other known index and are critical in public policy decisions. The default weight assigned to a set of select indicators is to assign equal weights without a precise weightage mechanism. The existing literature discussed earlier confirms the choice of these indicators.

By definition, these three are independent, although there may be intersectionalities. There is no evidence to show that caste, religion and domicile are correlated or have comparably different impacts on social decisions. This suggests that equal weights may be assumed and assigned to the components of the index. One of the features of the components is that the likelihood of multicollinearity is very low. Caste is not correlated to religion or domicile. But a system of caste may exist among major religions in India. Similarly, different castes will exist among those who are domiciled in a state as well as migrants.

Lines 283-286. These are repetitions of the previous lines. 

We thank the reviewer for pointing out this error. The text has now been updated with the deletion of the repeated text.

Lines 296-298. Please state briefly what the limitations are. 

We thank the reviewer for this comment and added text to mention the limitations briefly.

Lines 520-521. Centrally appointed administrators like the IAS, IPS are often posted in states other than the state of their own birth/domicile. As such, they would be much less likely to be influenced to grant favours to alleged criminals from the state (domicile) where these administrators are posted. Are there elements of political intervention? 

While it is true that centrally appointed officers are less likely to be influenced by local (state-level) hierarchies, two things need to be considered. First, many central officers seek deputations and appointments in their home states. This is a well-known phenomenon during cadre-level allocations. Second, the bulk of the governance process is handled by state administration and judiciary that is locally rooted and, therefore, likely to be influenced by pre-existing hierarchies. It is also true that even when central and state-level administrators act in a manner that does not give in to existing orders, the influence of political representatives is well-recognised. 

Line 571. The acronym BIMARU was a convenient way to describe the states which lagged behind other states in family planning and fertility decline and not exactly in terms of human development.

We have replaced all references in the text to BIMARU by EAG as suggested by the reviewer.

Lines 568 and onwards. Please see my previous comments about using BIMARU and replacing it with EAG states. 

Please see our response to the previous comment.

---

## [Editor Report · Decision Letter 2]

20 Jun 2023

What does the demographic profile of convicts tell us about social equity in India?

PONE-D-21-38262R2

Dear Dr..Pranab Mukhopadhyay,

We’re pleased to inform you that your manuscript has been judged scientifically suitable for publication and will be formally accepted for publication once it meets all outstanding technical requirements.

Kind regards,

Gouranga Lal Dasvarma, PhD

Academic Editor

PLOS ONE

Additional Editor Comments (optional):

Thank you for addressing my comments.
---

## [Editor Report · Acceptance letter]

29 Jun 2023

PONE-D-21-38262R2 

What does the demographic profile of convicts tell us about social equity in India? 

Dear Dr. Mukhopadhyay:

I'm pleased to inform you that your manuscript has been deemed suitable for publication in PLOS ONE. Congratulations! Your manuscript is now with our production department. 

Kind regards, 

on behalf of

Dr. Gouranga Lal Dasvarma 

Academic Editor

PLOS ONE